# Protection of Historical Mortars through Treatment with Suspensions of Nanoparticles

**Efstathia I. Pavlakou** [1,2], **Christine Lemonia** [1], **Emily Zouvani** [1], **Christakis A. Paraskeva** [1,2] and **Petros G. Koutsoukos** [2,3,*]

[1] Laboratory of Transport Phenomena and Physicochemical Hydrodynamics, Department of Chemical Engineering, University of Patras, 26500 Patras, Greece

[2] Institute of Chemical Engineering Sciences-Foundation of Research and Technology Hellas (FORTH-ICEHT), 26500 Patras, Greece

[3] Laboratory of Inorganic and Analytical Chemistry, Department of Chemical Engineering, University of Patras, 26500 Patras, Greece

[*] Correspondence: pgk@chemeng.upatras.gr

**Abstract:** Mortars, which are very important elements for the integrity of historic monuments, consist mainly of calcium carbonate and silicates in different proportions. Chemical dissolution due to exposure in open air is very important for the degradation of mortars. Inorganic nanoparticles with chemical and crystallographic affinity with mortar components are expected to be effective structure stabilizers and agents offering resistance to chemical dissolution. In the present work, we have developed and applied suspensions of amorphous calcium carbonate (ACC), silicon oxide (am-SiO$_2$) and composite nanoparticles by the precipitation of ACC on am-SiO$_2$ and vice versa. The application of suspensions of the synthesized nanoparticles on three different historical mortars of Roman times (1st century AD), retarded their dissolution rate in solutions undersaturated with respect to calcite, in acid pH (6.50, 25 °C). All three test historic mortars, treated with suspensions of the nanoparticles prepared, showed high resistance towards dissolution at pH 6.50. The ability of the nanoparticles' suspension to consolidate the damaged mortar was the key factor in deciding the corresponding effectiveness in the retardation of the rate of dissolution. The combination of ACC with am-SiO$_2$ nanoparticles showed high efficiency for protection from the dissolution of calcite rich mortars.

**Keywords:** mortar treatment; amorphous calcium carbonate; amorphous silica; nanoparticles; dissolution rate

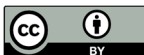

## 1. Introduction

The masonry of historic buildings is stabilized by the use of mortars, consisting of silica and calcite formed by the carbonation of Ca(OH)$_2$ [1]. The calcitic materials of the built cultural heritage deteriorate upon contact with water from wet precipitation [2,3]. Calcite is the basic component and model inorganic compound of calcitic building elements and is susceptible to the action of various physical, chemical and biological factors [4]. SO$_2$, NO$_x$ and volatile organic carbon (VOC) gases in the atmosphere, temperature, light, the presence of microorganisms on the surface of the monuments and human interventions contribute to materials' erosion over time [5–7]. Given the value and importance of cultural heritage, the restoration of damage and conservation is urgent. Progress in nanotechnology provided novel methods and materials for the restoration of cultural heritage monuments and artefacts [7,8]. The use of Ca(OH)$_2$ and Ba(OH)$_2$ nanoparticles has been shown to be effective in the restoration of works of art [5,9]. Microemulsions, micellar solutions and cleaning gels have also been successfully used for the restoration of artefacts [5,10–12]. Polymer chemical coatings have been shown to protect limestone, but outdoor exposure is destructive due to the ageing of the polymeric materials [13,14]. As

shown over the last two decades, nanoparticles of metal oxides and hydroxides ($TiO_2$, ZnO, $Ca(OH)_2$, $Mg(OH)_2$, $Sr(OH)_2$, and metal nanoparticles (Au, Ag, Pt) have been successfully tested as consolidants and/or protective additives on different works of art [15–17]. Colloidal silica particles have been successfully used in the restoration and conservation of cultural heritage stone monuments, protecting them from further deterioration, and they contributed to their consolidation [18]. Silica nanoparticles protect calcareous materials penetrating inside pores in which their contact with calcium carbonate material results in rapid agglomeration and the development of a protective layer [19]. Using colloidal silica, suspended in aqueous systems or in water containing ethanol, hydroxyl groups on the surface of the particles condense upon drying, by the removal of water to form siloxane bonds (Si-O-Si), resulting in coalescence and inter-bonding [20]. The application of silica nanoparticles, average size 10–15 nm, has also been reported in mortars and it was concluded that they interact with calcium carbonate forming xerogels upon water loss [21]. The treatment of Tuff stone with suspensions of colloidal silica particles provided satisfactory protection, filling pores and penetrating the material in depth [22]. Although not well understood, there are significant interactions between silica and silicic species with calcium carbonate, which result to the consolidation of the latter [23]. The problem with silica colloidal particles formed by the hydrolysis of TEOS is that the layers formed inside the pores, or on surfaces, tend to crack [24].

Amorphous calcium carbonate (ACC) is a type of calcium carbonate that has a disordered, non-crystalline structure. ACC particles have been shown to be efficient consolidants for calcareous matrices, suggesting that ACC has a high potential for protection of stone cultural relics with calcareous matrix [25]. Nanolime particles have been applied for calcareous stone protection that involve the conversion of calcium hydroxide- $CO_2$ reaction products into ACC [26,27]. ACC and calcium oxalate particles in suspension applied on white marble, calcarenite limestone and gypsum plaster, showed enhanced hydrophobicity, less cracking, improved surface adhesion, and a marked improvement in acid resistance [28]. ACC is often used as a protective coating for stone surfaces to help protect against weathering. The use of ACC as a protective coating for stone surfaces is based on the fact that calcium carbonate is a relatively stable and durable material. When applied to a stone surface, ACC can help to form a protective layer, and is able to resist the effects of weathering, such as water and wind erosion, and UV radiation. ACC layers can help to preserve the appearance and structural integrity of the stone over time. ACC coating can be applied with a variety of methods, including spraying, brushing and dipping, to protect historic and cultural stone buildings, statues and other structures both indoors and outdoors. [29–31]. Several factors may affect the effectiveness of ACC as a protective coating, including the type of the treated material, the climate, and the type of weathering. In general, ACC is most effective at protecting against water and wind erosion but may not be as effective at protecting against other types of weathering, such as chemical attack or physical abrasion. Silica stabilization has been shown to improve the physical and chemical stability of ACC, making it less prone to dissolution and more resistant to changes in pH and temperature [32]. ACC may from directly on silica particles with methanol playing a stabilizing role for ACC [33]. Mortars are mixtures of inorganic or organic binders, predominantly fine aggregates, water and in some cases organic and/or inorganic additives (or mixture of binder only and water). They are mixed in appropriate proportions to provide mixture, in the fresh state, a suitable workability and, in the hardened state, adequate physical characteristics (porosity, water permeability etc.), mechanical characteristics (strength, deformability, adhesion, etc.) appearance, and durability. The evidence from the literature reports that the effectiveness of treatment of materials with nanoparticles depends on the extent of interactions with the matrix materials. Since mortars contain silica and calcium carbonate, both silica and calcium carbonate particles are expected to show compatibility. The preparation of nanoparticles consisting from ACC core and am-$SiO_2$ coverage and am-$SiO_2$ core with ACC over layering aimed at the treatment of composite materials, like the mortars, with chemically and crystallographically compatible

components. In addition, with the information provided by literature reports on a broad range metal oxides nanoparticle, in the present study we have focused on nanoparticles compatible with the components of the treated materials using all possible combinations of ACC and am-SiO$_2$ nanoparticles. Historical mortars were used for treatment and they were tested with respect to their resistance in chemical weathering. The tests were based on the measurement of the rates of dissolution of the treated mortars, and of the nanoparticles used for the treatment, independently, in solutions undersaturated with respect to calcite at constant pH, which simulate better wet precipitation.

In the present work, the protective effect of am-SiO$_2$, ACC, (am-SiO$_2$)-ACC (composite 1, SY1) and (ACC)-am-SiO$_2$ (composite 2, SY2) nanoparticles was tested on mortars. Three historical mortars, consisting mainly of calcite and silica with various CaCO$_3$/SiO$_2$ proportions, were treated with suspensions of the nanoparticles. All specimens were treated in powdered form to maximize the exposed total surface area exposed in the weathering medium. The protection of the treated specimens from deterioration was evaluated by measurements of the rates of dissolution in undersaturated conditions of calcium carbonate (pH 6.50, 25.0 °C), and the effectiveness of the treatment materials and methods of application were evaluated. The composition of the undersaturated solutions containing calcium, carbonate, sodium and chloride ions simulated wet precipitation. [32]. The selected pH value was in the acid side and not very acid as in the case of heavily polluted atmosphere from acid gases (SO$_x$, NO$_x$).

## 2. Materials and Methods

Historical mortar samples were used for the evaluation of the nanoparticles tested with respect to their ability to protect them from chemical weathering in acid solutions. Three historical mortars obtained from Roman stadium in Patras (built between 80–90 AD, probably a gift from the Roman emperor Domitian, Patras, Greece) with different CaCO$_3$/SiO$_2$ proportions, were treated with suspensions of nanoparticles. The specific surface area and the porosity of the test materials were measured by nitrogen adsorption using the BET method (Micromeritics, Gemini 2370) (Table 1). The mortar specimens were treated with nanoparticles suspending 1 g of powdered material in nanoparticle suspensions formed as described in detail, next. The nanoparticles used in the treatment of mortars were ACC, am-SiO$_2$ and combinations of ACC and am-SiO$_2$, Composite 1 (SY1), Composite 2 (SY2)). All syntheses were done in a double walled borosilicate glass reactor, thermostated at 25.0 ± 0.5 °C, magnetically stirred to ensure homogeneity.

**Table 1.** Physico-chemical characteristics of test materials before treatment with suspensions of nanoparticles. (* the rest is silica and aluminosilicates)**.**

| Material Treated | % Ca * | Specific Surface Area (m$^2$/g) | Porosity (cm$^3$/g) |
|---|---|---|---|
| Mortar 1 | 61 | 10.0 | 0.021 |
| Mortar 2 | 50 | 13.2 | 0.022 |
| Mortar 3 | 43 | 14.0 | 0.064 |

### 2.1. Synthesis of Nanoparticles

ACC was prepared by the addition of 0.05 M dimethyl carbonate (DMC, Merck SA, For synthesis, Taufkirchen, Germany), sodium chloride (NaCl, Merck Greece SA, Athens, Greece) 0.1 M and calcium chloride (CaCl$_2$, Riedel de Haen, Seelze, Germany) 0.01 M in a 150 mL, magnetically stirred batch reactor (Figure 1) at 25.0 °C (concentrations correspond to the final solution prepared) [34]. Mixing of the solutions lasted 2.5 min before the separation of the precipitate from the solution. The solid was separated from the suspension by centrifugation at 2500 rpm for 4 min, then by filtration through membrane filters (0.20 μm, Sartorius, Goettingen, Germany) and it was rinsed with acetone and freeze dried. am-SiO$_2$ was prepared by the addition of tetraethyl orthosilicate (TEOS, Sigma Aldrich,

Merck, Darmstadt, Germany) (0,29 M) solution, in a solution of 11 mL ethanol ($C_2H_5OH$, 95% *w/v*) and 5 mL ammonia ($NH_3$, 32% *w/v*). Before the addition of TEOS, the ethanol-ammonia solvent mixture was homogenized by magnetic stirring for 30 min TEOS hydrolysis started immediately upon addition in the ethanol-ammonia solution, resulting to the formation of am-$SiO_2$. The suspension was stirred for 1 h from the start of TEOS addition. [35]. Next, the solid was separated from the suspension by centrifugation at 40,000, rpm for 30 min. Composite nanoparticles SY1 were prepared by the addition of a 0.02 g of am-$SiO2$ in a batch reactor (150 mL) containing DMC, NaCl and $CaCl_2$ solutions. The pH was adjusted to 10 by the addition of NaOH as needed (20 mL of 0,5 M NaOH solution) (Figure 1). The suspension was homogenized by stirring with a magnetic stirrer for 2.5 min [35]. SY1 particles were separated by filtration was rinsed with acetone and freeze dried. SY2 nanoparticles, were prepared by the addition of ACC nanoparticles to a suspension of $SiO_2$ nanoparticles prepared in situ. Specifically, 0.07 g of ACC nanoparticles were introduced in a reactor, containing 115 mL of 0.67 M ammonia solution in 95% ethanol under stirring. Next, TEOS (0.29 M final concentration) was added and stirring continued for 10 min (Figure 1) [36].

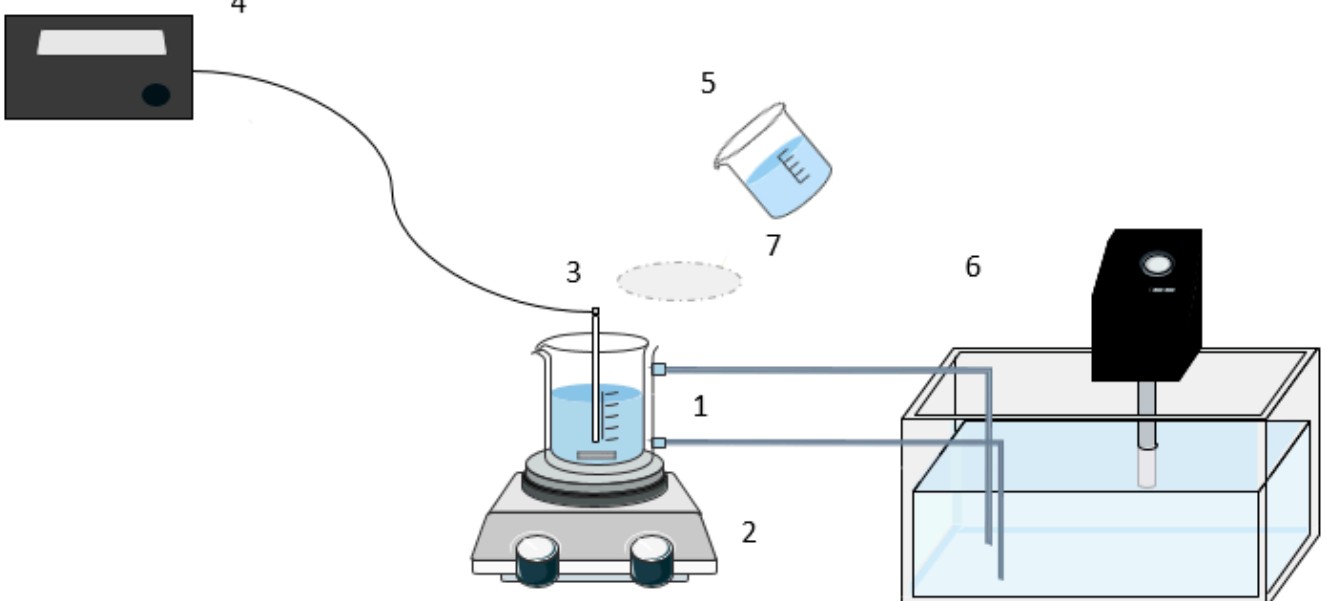

**Figure 1.** Experimental setup of the synthesis of nanoparticles (ACC, am-$SiO_2$, SY1 and SY2) and for the treatment of three historical mortars: 1: thermostated batch reactor; 2: magnetic stirrer; 3: glass/Ag/AgCl combination electrode; 4: pH meter; 5: added solutions; 6: bath; 7: polyamide reactor cap.

*2.2. Treatment of the Test Specimens with the Nanoparticles*

ACC, am-$SiO_2$, SY1, SY2 were used for the treatment of the test samples (Table 1). The treatment was carried out in a thermostated double walled borosilicate glass reactor, volume totaling 150 mL, at 25 °C. The suspensions in the reactor were homogenized by magnetic stirring using a Teflon® coated stirring bar (Figure 1). 1 g of the test sample specimens was suspended in a solution containing 0.05 M DMC and 0.01 M $CaCl_2$ solutions. The suspension of the test specimen particles was mixed thoroughly until the suspension was homogeneous. Next, 0.1 M standard sodium hydroxide solutions was added to adjust pH ≈ 10. At this alkaline solution DMC was hydrolyzed [37]. The release of carbonate ions, $CO_3^{2-}$, in combination with the presence of free calcium ions developed the supersaturation needed for the spontaneous formation of ACC:

$$(CH_3)_2CO_3 + 2NaOH \rightarrow 2CH_3OH + Na_2CO_3 \tag{1}$$

This treatment resulted in the formation of ACC, which was deposited, at least partly, on the suspended particles of the treated materials (three historic mortars). The methanol released during the hydrolysis of dimethyl carbonate has a stabilizing effect on the ACC [25]. In an analogous way, the specimens to be treated were suspended in a solution in which am-$SiO_2$ was precipitated. 1 g of the powdered test specimens (Table 1) was suspended in the 0.67 M ammonia solution in 95% ethanol. Next, TEOS was added to a final concentration of 0.29 M. Am-$SiO_2$ takes place in this solution by successive hydrolysis of TEOS and condensation reactions as shown in Figure 2:

**Figure 2.** Synthesis of am-$SiO_2$ by hydrolysis of TEOS. Condensation reactions.

Ethoxy ortho silicic acd and orthosilicic acid are produced by the partial and full hydrolysis Equations (1) and (2) shown in Figure 2. Next, these two products are condensed according to Equation (3) forming disiloxane. At subsequent reaction (Equation (4)) trisiloxane and polysiloxanes are produced, until a critical size is reached characteristic of the am-$SiO_2$ precipitated.

Treatment with the composite particles SY1 and SY2 was carried out by the addition of 1.0 g of the specimens for treatment in suspensions of the corresponding particles under stirring. The contact time was limited to 10 min. Following suspension, SY1 and SY2 nanoparticles were separated from the liquid phase by filtration through membrane filters. The solid was freeze dried. The morphology of the nanoparticles and of the test materials were examined by scanning electron microscopy (SEM, Leo Supra 35 VP, Zeiss, Oberkohen, Germany). The treatment of the mortars with SY1 and SY2 did not change significantly the composition of the material tested by dissolution in the first two cases. The composition of mortar 1 was 61% *w/w* $CaCO_3$ and 39% *w/w* $SiO_2$. After treatment with SY1,

the concentration for SiO₂ was 39% *w/w* and with SY2, the concentration of SiO₂ was 37% *w/w*. The composition of mortar 2 was 50% *w/w* CaCO₃ and 50% *w/w* SiO₂. After treatment with SY1, SiO₂ was 49% *w/w* and with SY2, SiO₂ was 47% *w/w*. The composition of mortar 3 was 43% *w/w* CaCO₃ and 57% *w/w* SiO₂. After treatment with SY1, the concentrations the concentration of SiO₂ was 42% *w/w* and with SY2, SiO₂ was 40% *w/w*.

### 2.3. Dissolution Tests

The dissolution of the test specimens was studied in undersaturated calcium carbonate solutions at constant pH. In all cases, the calcium concentration was monitored during dissolution, in undersaturated solutions with respect to calcite, the thermodynamically most stable polymorph of calcium carbonate. Monitoring was performed by sampling and analysis of the samples for their calcium content in the aqueous phase during dissolution. The undersaturated solutions were prepared in a thermostated batch reactor mixing equal volumes of standard CaCl₂.2H₂O and NaHCO₃ solutions, 75 mL each. Both solutions were prepared from the respective stock solutions, prepared from the corresponding crystalline materials. The calcium chloride stock solutions were standardized with atomic absorption spectrometry (AAS, Perkin Elmer AAnalyst 300) and by titration with standard EDTA solutions with murexide indicator. The sodium bicarbonate stock solutions were prepared fresh in every experiment from the respective crystalline solid (Merck Puriss) dried overnight at 70 °C and used without any further standardization. The ionic strength of the solutions was adjusted with sodium chloride from the respective stock solution, made from the solid without any further standardization. In all measurements the total calcium concentration, ($Ca_t$), in the undersaturated solutions was equal to the total carbonate concentration ($C_t$) and equal to 1.25 mM. The ionic strength was 0.1 M in NaCl. The pH of the undersaturated solutions was acid (6.50) and it was adjusted by the addition of standard HCl solution (Merck, titrisol). Following the verification of the constancy of the solution pH, 10 mg of the test material was introduced in the undersaturated solutions and it was rapidly dispersed by magnetic stirring. Throughout the dissolution of the calcium carbonate component of the test specimens the pH of the undersaturated solutions was kept constant by the addition of standard hydrochloric acid to neutralize the protons released:

$$CaCO_3(s) + H_2O \leftrightarrows Ca^{2+}(aq) + HCO_3^-(aq) + OH^-(aq) \tag{5}$$

Increase of the solution pH by 0.005 pH units, caused by the dissolution of the carbonate component of the dissolving solid (Equation (5)), triggered the addition of standard hydrochloric acid solution from a motorized syringe controlled by a digital computer (Figure 3). During dissolution, samples were withdrawn, filtered through membrane filters and the filtrates were analyzed for (total) calcium by AAS. Analysis for total dissolved silicate was done spectrophotometrically at 350 nm (UV-VIS spectrophotometer Perkin Elmer Lambda) [38] by the construction of the appropriate reference curve of the absorbance of standard samples as a function of their concentration. The analytical method was based on the complexation of silicate ions with ammonium molybdate. The initial rates of dissolution were calculated from the calcium-time profiles and were expressed per unit surface area of the dissolving solid.

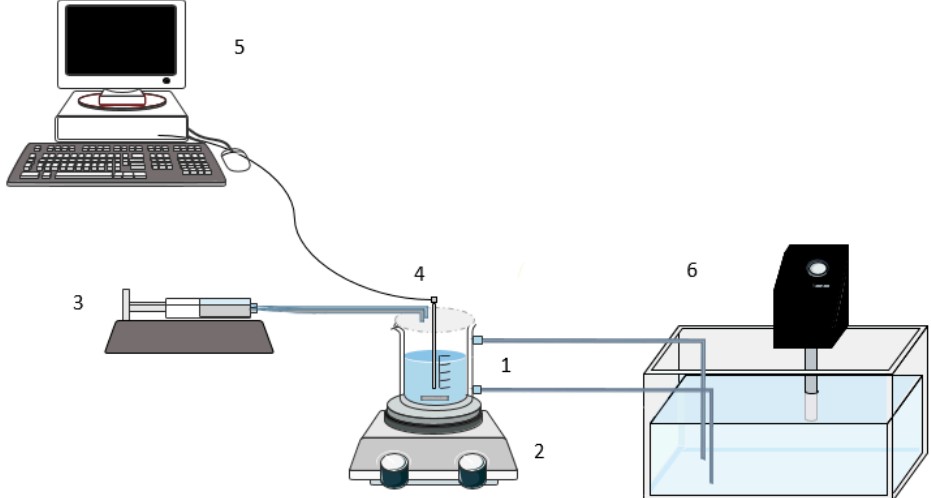

**Figure 3.** Experimental setup of the dissolution of the calcitic materials: 1: thermostated batch reactor; 2: magnetic stirrer; 3: motor driven syringes; 4: glass/Ag/AgCl combination electrode; 5: digital computer (data collection, syringe control); 6: bath.

## 3. Results

The deterioration of untreated and treated mortar samples was examined by measurements of the kinetics of dissolution, which was assessed by the rate of calcium increase in the solutions as a function of time and from the examination of the morphology of the dissolving material (before and after treatment with the nanoparticles). The morphology of the three mortars is shown in the SEM pictures in Figures 4a-c, respectively. In the first two mortars (60% and 50% calcite and the rest silica), the larger calcite crystals were partially covered by silica nanoparticles. In mortar 3 (Figure 4c) silica was in excess (60%) and the coverage of the calcitic particles was denser as may be seen from the texture of the mortar particles. The mortars were treated with amorphous calcium carbonate, (ACC), amorphous silica (am-SiO$_2$), SY1 and SY2 nanoparticles. Nanoparticles SY1 and SY2, consisted of ACC and am-SiO$_2$ in different proportions. The different calcium carbonate -silica content was due to the fact that in the case of SY1, the nanoparticles were prepared by a core of am- SiO$_2$ covered by ACC and SY2 by a core of ACC nanoparticles covered by the much smaller am-SiO$_2$. During the preparation of SY1, the silica nanoparticles, which were initially introduced into the synthesis solution, created a layer, on the surface of which a second layer, consisting of ACC nanoparticles was deposited (Figure 5a). Accordingly, SY2 was formed when a layer of silica nanoparticles was deposited on top of ACC nanoparticles (Figure 5b). As may be seen, the larger number of the am-SiO$_2$ nanoparticles resulted in the formation of an outer layer of ACC, but the presence of am-SiO$_2$ was also significant. The composition of SY1 was calculated as 69% ACC/31% am-SiO$_2$ while that of SY2 was 97% ACC/3% am-SiO$_2$.

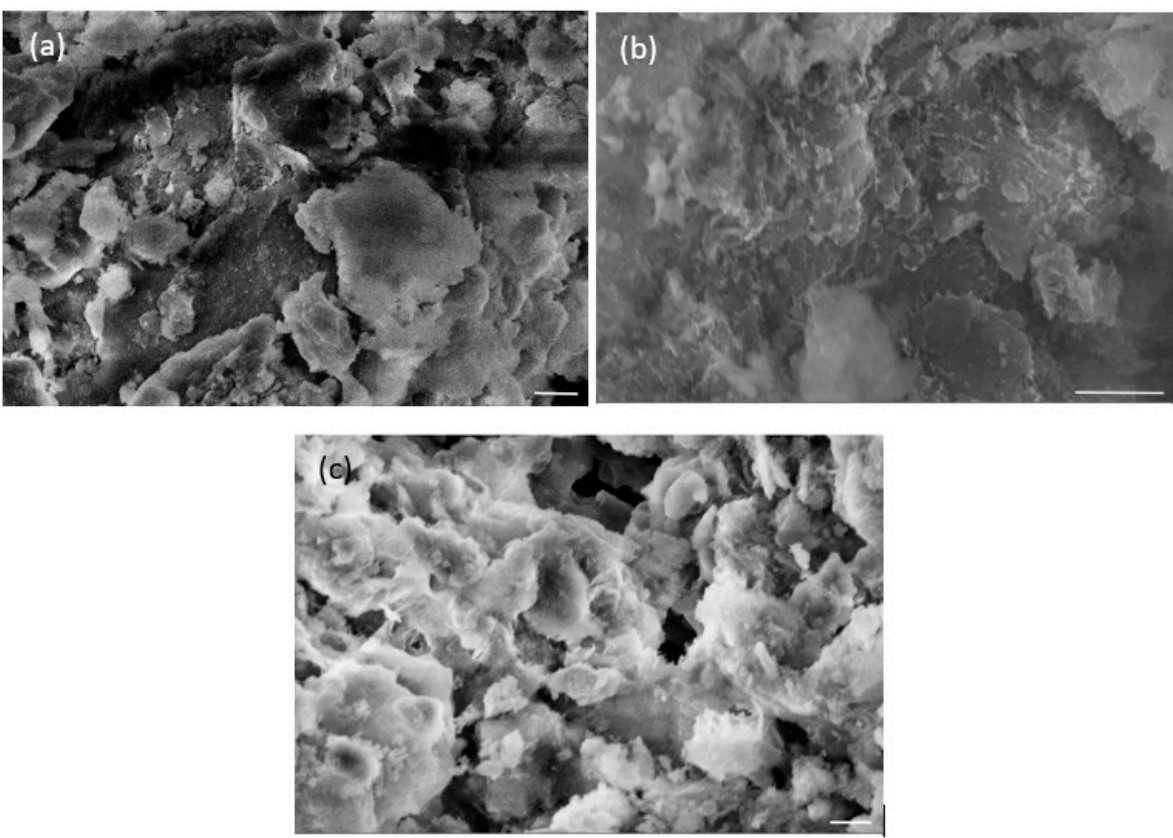

**Figure 4.** (**a**) Mortar 1 (bar = 2 μm), (**b**) mortar 2 (bar = 1 μm), (**c**) mortar 3 (bar = 2 μm).

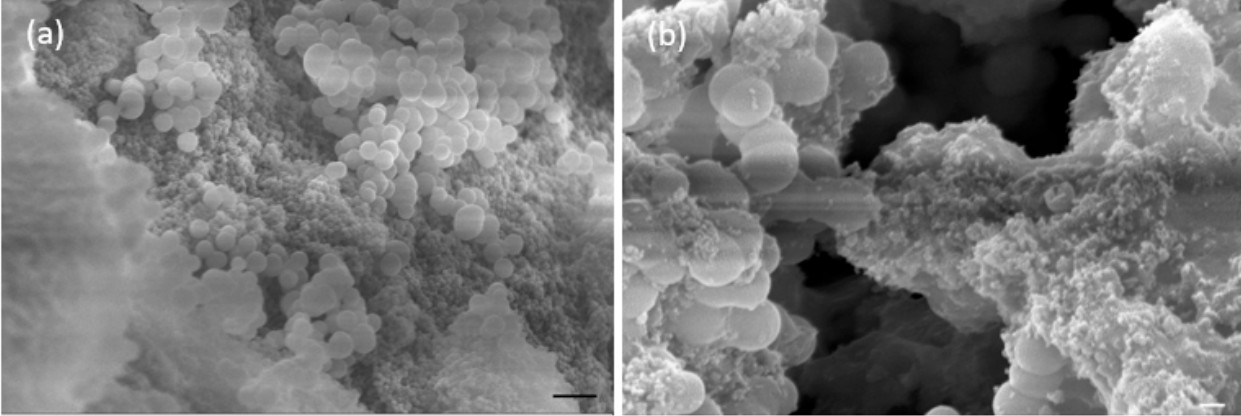

**Figure 5.** (**a**) SY1, bar = 1 μm; (**b**) SY2, bar = 200 nm.

The coating of mortar_1 with ACC nanoparticles is shown in Figure 6a. In some cases, the transformation of ACC to calcite is evident. The particle size of the deposits is in the nanometer range (ca. 450 nm), which is due to the initial formation of ACC. The coverage of the mortar grains appears to be complete. The chemical and crystallographic affinity of calcium carbonate and silica was the key factor for the coverage of the mortar surface [39]. The precipitated silica nanoparticles (≈70 nm) formed a layer covering completely the surface of the treated mortar as shown in Figure 6b. In Figure 6c the grains of mortar_1, treated with SY1 nanoparticles are shown. The amorphous silica nanoparticles, acting as substrate, penetrated the pores of the treated mortar. ACC nanoparticles precipitated on

the template provided by the silica nanoparticles, reinforced the coating of the mortar grains. In Figure 6d the result of treatment of the mortar sample 2 with SY2 nanoparticles is shown. The coverage is excellent and as shown the silica nanoparticles covered the first layer of the underlying ACC nanoparticles, which are in direct contact with the mortar grains.

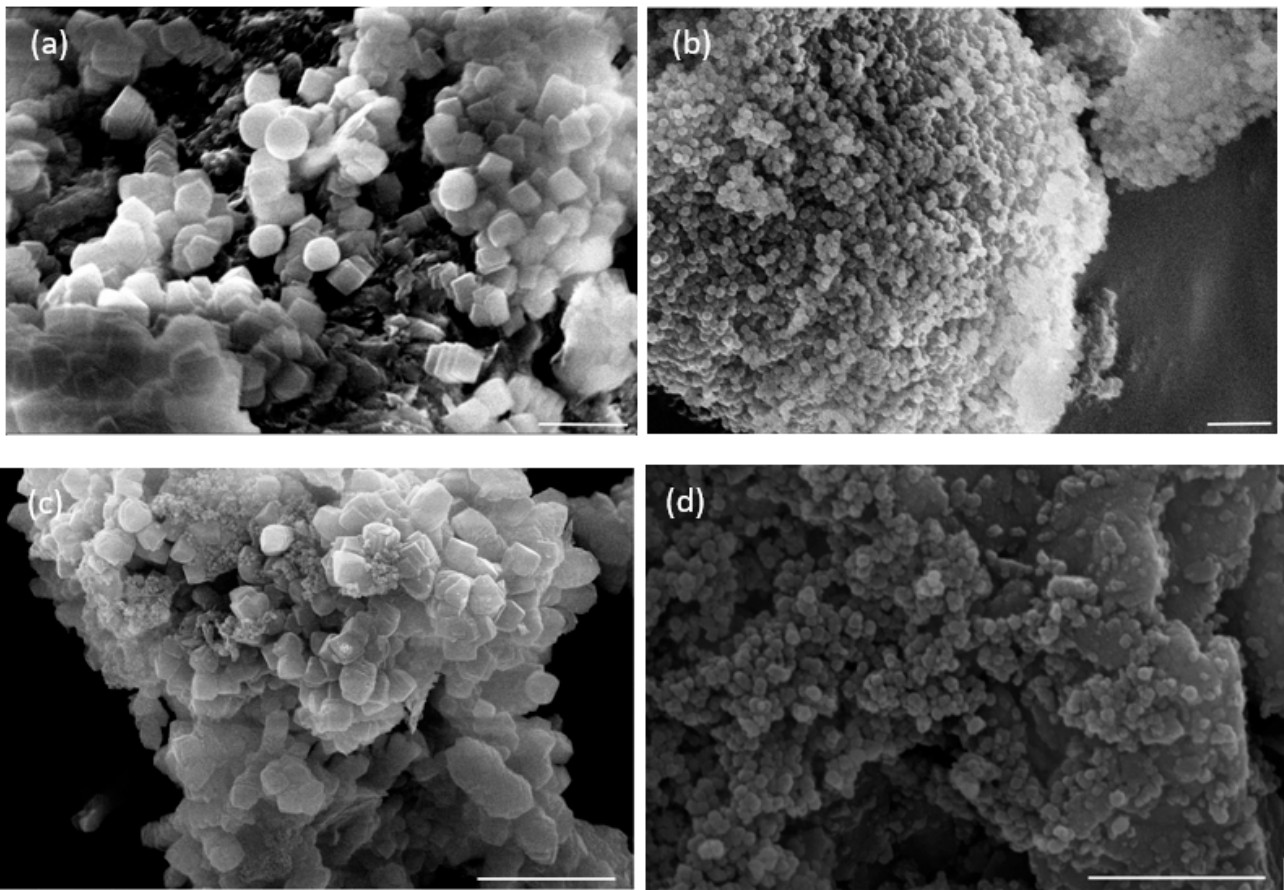

**Figure 6.** Mortar_1 with deposits (**a**) ACC deposits, bar = 1 μm; (**b**) SiO₂ deposits, bar = 1 μm; (**c**) SY1 deposits, bar = 5 μm; (**d**) SY2 deposits, bar = 1 μm.

In the SEM pictures shown in Figure 7, ACC crystals deposited on the surface of mortar sample 2, started transforming to the thermodynamically most stable calcium carbonate, rhombohedral calcite (Figure 7a). Upon coverage with am-SiO₂ (Figure 7b), it is very likely that the deposition of the respective nanoparticles took place not only on ACC but also inside the pores of the mortar sample. In the deposition of SY1 nanoparticles on the grains of mortar 2, ACC is still shown together with the calcite crystals from the transformation of ACC (Figure 7c). According to the treatment methodology with SY1 nanoparticles, am-SiO₂ covered the surface and the pores of mortar sample 2, followed by the deposition of ACC. In the treatment of the mortar samples with SY2 nanoparticles, ACC deposited first by spontaneous precipitation, followed by the deposition of am-SiO₂ by TEOS hydrolysis, which resulted in the rapid precipitation of the respective nanoparticles, as shown in the SEM picture presented in Figure 7d.

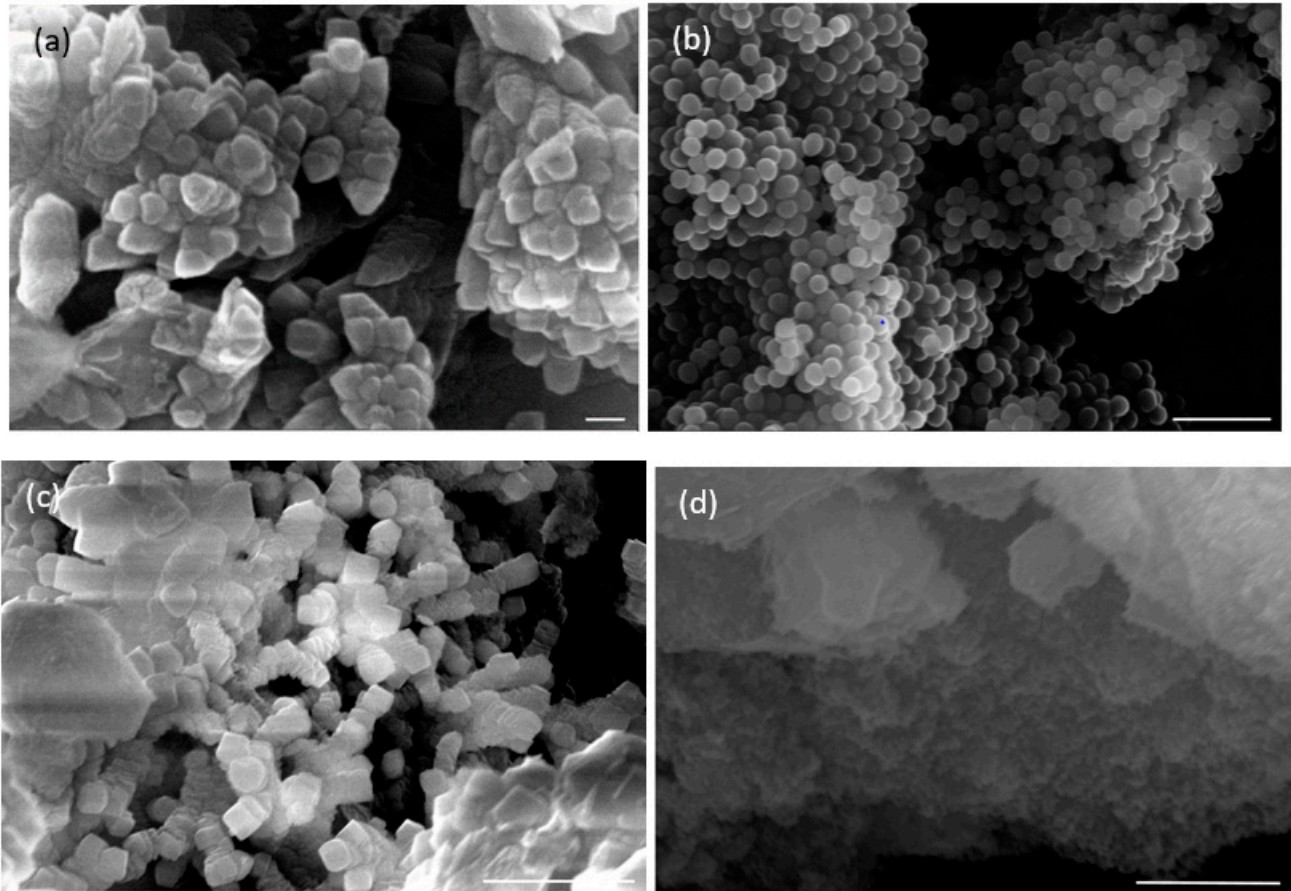

**Figure 7.** Mortar_2 treated with nanoparticles (**a**) ACC, bar = 200 nm; (**b**) am-SiO$_2$, bar = 1 $\mu$m; (**c**) SY1, bar = 3 $\mu$m; (**d**) SY2, bar = 2 $\mu$m.

In the case of treatment of the Si-rich mortar sample (mortar sample 3), as may be seen in the SEM picture shown in Figure 8a, ACC nanocrystals covered efficiently the mortar.

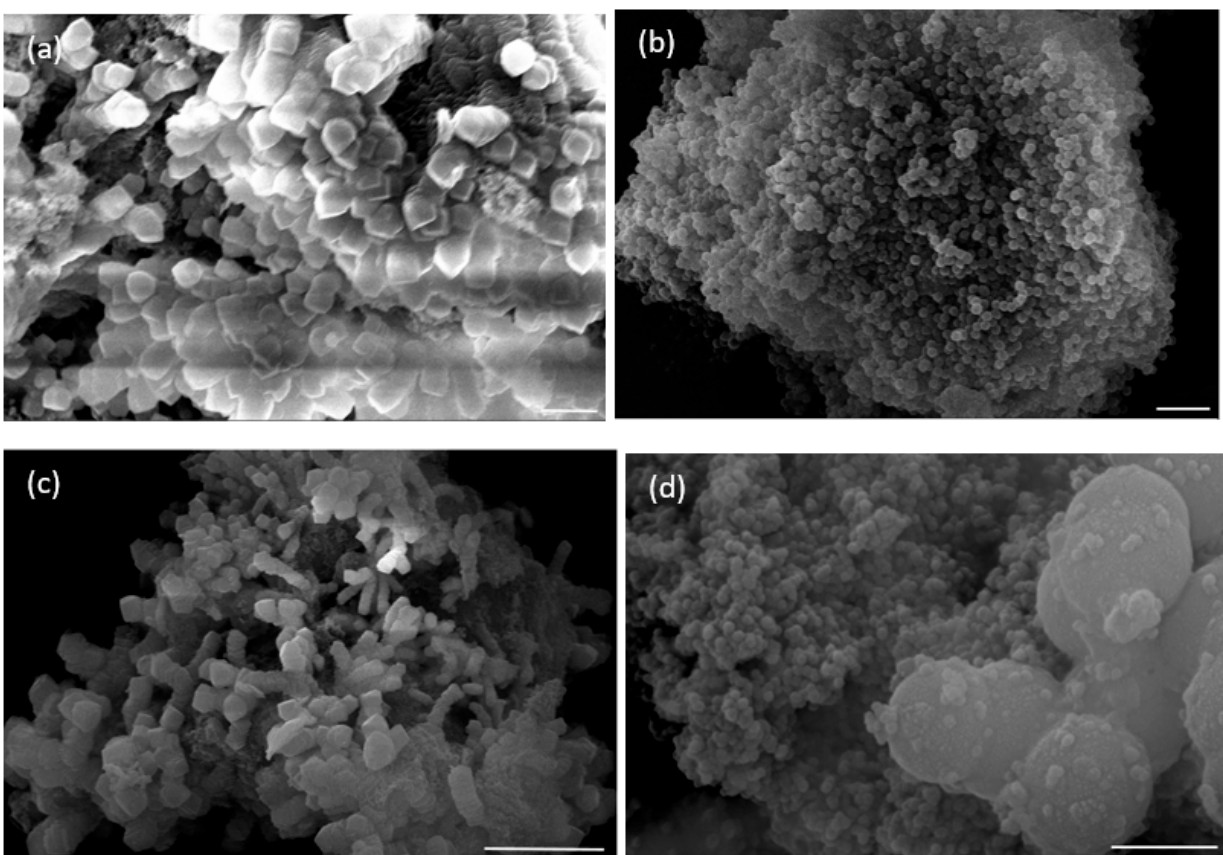

**Figure 8.** Mortar sample 3 treated with nanoparticles: (**a**) ACC, bar = 1 $\mu$m; (**b**) am-SiO$_2$, bar = 1 $\mu$m; (**c**) SY1, bar = 5 $\mu$m; (**d**) SY2, bar = 500 nm.

### 3.1. Composites of Silica Nanoparticles with ACC and Silica Particles

These methods involve the formation of a stable dispersion of ACC and silica particles, which can then be dried to form a solid composite material. Silica stabilization has been shown to improve the physical and chemical stability of ACC, making it less prone to dissolution and more resistant to changes in pH and temperature. ACC has been suggested to be stabilized by the presence of the large tetrahedral silicate ions destabilize calcite [40]. Moreover, silica particles in the presence of NaCl have been reported to enhance attachment to calcium carbonate particles [41]. Am-SiO$_2$ deposited on mortar sample 3 (Figure 8b), formed a uniform rather compact overlayer. In the SEM shown in Figure 8c, ACC nanoparticles layers on the mortar are visible. Some transformation to the thermodynamically more stable calcite is seen. In the case of treatment with SY1 nanoparticles, silica particles deposited served as a substrate for ACC over layering. The fact that in the mortar specimen 3, the silica content was in excess, favored the formation of the am-SiO$_2$ layer because of chemical affinity. In this case, the silica component acted as seed crystals, eliminating the nucleation barrier for am-SiO$_2$. In the treatment of the mortar sample 3 with SY2 nanoparticles, the silica nanoparticles that have settled on the larger ACC nanoparticles can be seen. The latter cover the mortar grains well, as shown in the SEM in Figure 8d.

### 3.2. Kinetics of Dissolution

Materials containing calcium carbonate dissolve in solutions undersaturated with respect to calcite, the thermodynamically most stable calcium carbonate polymorphs, because of the dissolution of the calcium carbonate component. The saturation of calcium carbonate solution with respect to calcite, S$_{calcite}$, is given by Equation (6):

$$S_{calcite} = \frac{(Ca^{2+})(CO_3^{2-})}{K_{s,calcite}^0} = \frac{(IAP)_{calcite}}{K_s^0} \tag{6}$$

In Equation (6), ( ) denote the activities of the ions inside parentheses, their product being the ion activity product, IAP, and $K^0_{s,\,calcite}$ is the thermodynamic solubility product of calcite. For $S_{calcite} = 1$ the solution is saturated with respect to calcite and for $S_{calcite} < 1$ it is undersaturated. The relative undersaturation with respect to calcite is defined as:

$$\sigma_{calcite} = 1 - S_{calcite}^{1/2} \tag{72}$$

The rate of dissolution, $R_{diss}$, of any calcitic material depends on the solution undersaturation:

$$R_{diss} = k_{diss}\sigma_{calcite}^n \tag{8}$$

In Equation (8) $k_{diss}$ is the apparent rate constant of dissolution and n the apparent order of the process. In this work, the results reported refer to $\sigma_{calcite}$ = 0.89. According to Equation (7), the relative undersaturation takes values in the range $0 < \sigma_{calcite} < 1$. The dissolution of mortar sample 1 (61% Ca/39% Si) in solutions undersaturated with respect to calcium carbonate, at constant pH 6.50, resulted in the increase of the calcium concentration in the undersaturated solutions as a function of time. At the initial stages the rate of calcium concentration change was rapid, slowing down with increasing calcium concentration in the solutions due to the dissolution of the carbonate content of the mortars. This results to decreasing relative undersaturation ($\sigma_{calcite} \rightarrow 0$) and to lower rates of dissolution. The concentration–time profiles of calcium and dissolved silicate, respectively, are shown in Figure 9a,b. The dissolution rates of the calcitic component of the mortars both untreated and treated with the methods described, were initial rates, calculated according to Equation (9):

$$R_{diss} = \left.\frac{d[Ca_t]}{dt}\right|_{t \rightarrow 0} \tag{9}$$

The initial rates of dissolution were calculated from the polynomial fit of the total calcium ($Ca_t$)–time profiles of each of the test samples [42]. The numerator in Equation (6) yielded the amount of calcium carbonate dissolved, as dictated by the stoichiometry of the dissolution Equation (5). The values calculated for the rate of dissolution for all mortar samples without and after treatment with the nanoparticles described are summarized in Table 2. The dissolution rate of mortar sample 1, treated with ACC nanoparticles, was lower in comparison to the respective rate of the untreated mortar, even though the rate of dissolution of ACC nanoparticles was higher in comparison to the respective untreated mortar sample. This reflects the intraparticle affinity between the mortar material and the nanoparticles, possibly with more contribution by the silicate component of it. The deposition of silica nanoparticles yielded dissolution rate for the treated mortar sample 1, one order of magnitude lower, in comparison with the respective rate obtained for the untreated sample. The treatment of mortar sample 1 with SY1 and SY2 nanoparticles yielded lower dissolution rates in comparison with the untreated mortar sample 1. Treatment with SY2 nanoparticles yielded lower dissolution rates in comparison with SY1, possibly because the effect of am-$SiO_2$ which sealed the pores and covered the surface of the mortar grains. ACC formed on the am-$SiO_2$ inlay. The ACC overlayer was stabilized by interactions with the am-$SiO_2$.

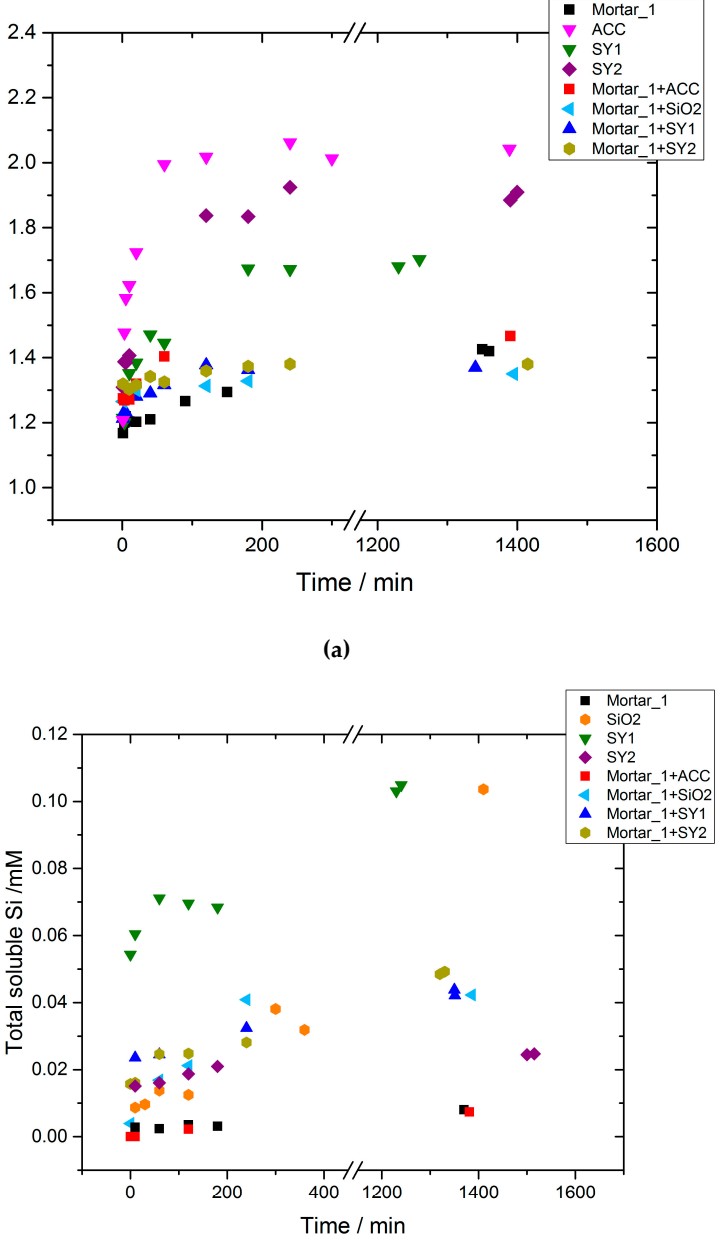

**(a)**

**(b)**

**Figure 9. (a)** Total calcium - timeprofile for the dissolution of mortar sample 1 in undersaturated calcium carbonate solutions ($\sigma$ = 0.89) at constant pH; 25 °C; pH 6.50; 0.1 M NaCl; (■)Mortar 1; (■) treated with ACC nanoparticles; (◄) treated with SiO$_2$ nanoparticles; (▲) treated with SY1 nanoparticles; (●) treated with SY2nanoparticles; **(b)** total dissolved Si-time profiles. (■) Mortar 1; (●) am-SiO$_2$; (▼) SY1; (◆) SY2;(■)Mortar 1+ACC; (◄) Mortar 1+am-SiO$_2$; (▲) Mortar 1+SY1; (●) Mortar 1+SY2.

The total calcium and total silicate profiles during the dissolution of mortar sample 2 (50% Ca/50% Si) in solutions undersaturated with respect to calcium carbonate are presented in Figure 10, in which the total calcium and total silicate- time profiles are shown. The rate of dissolution of the mortar did not decrease significantly past treatment with ACC nanoparticles. It should be noted again that the rate of dissolution of ACC nanoparticles measured at the same conditions, was significantly higher than the corresponding rate of dissolution of the mortar. Treatment with am-SiO$_2$ nanoparticles reduced the

dissolution rate of the mortar in comparison with the untreated mortar. The reduction in the rate was significant but not as high as in the case of mortar sample 1. It should be noted that in this case the dissolution of the coating contributed to the silicate concentrations in the undersaturated solutions, as shown in the silicate -time profile (Figure 10). The treatment of the mortar with SY1 and SY2 nanoparticles did not have a significant effect, but it is important to note that both calcium and silicate released in the solutions were lower in comparison with the corresponding free ACC and am-$SiO_2$ nanoparticles. This revealed the presence of **a** strong interaction between the treatment nanoparticles and the mortar treated. From the morphology examination it was clear that the relatively higher levels of calcium and silicate in the undersaturated solutions were due to partial dissolution of the coating nanoparticles, which provide protection to the mortar against dissolution through sacrificial dissolution, protecting the original material. This suggestion is due to the fact that the nanoparticles from the treatment cover the outer surface of the mortar and dissolution takes place at the outer surface of the treated solid.

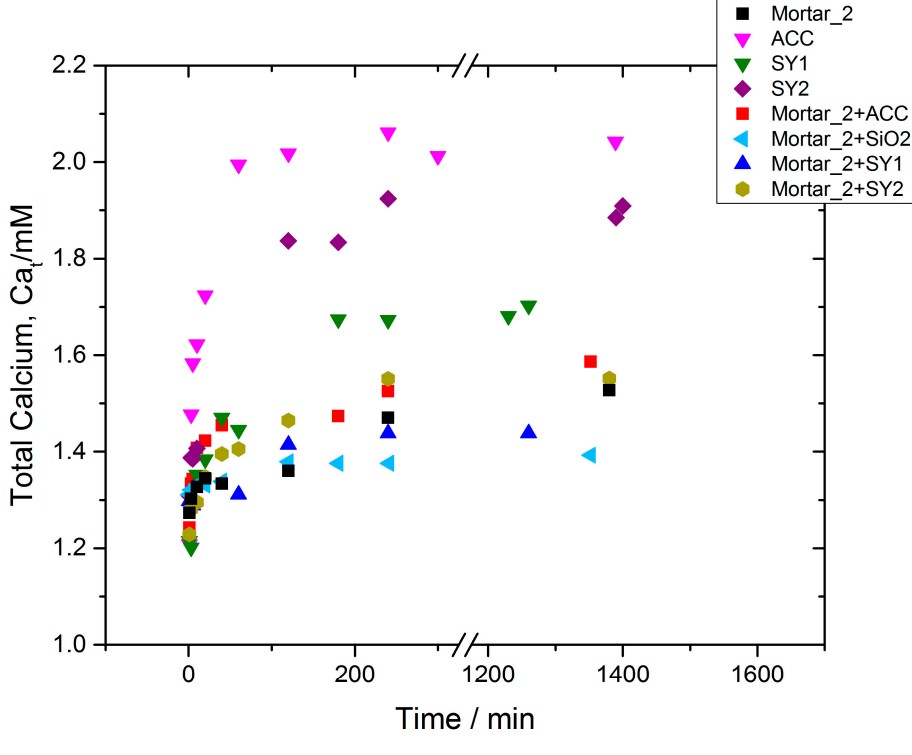

**(a)**

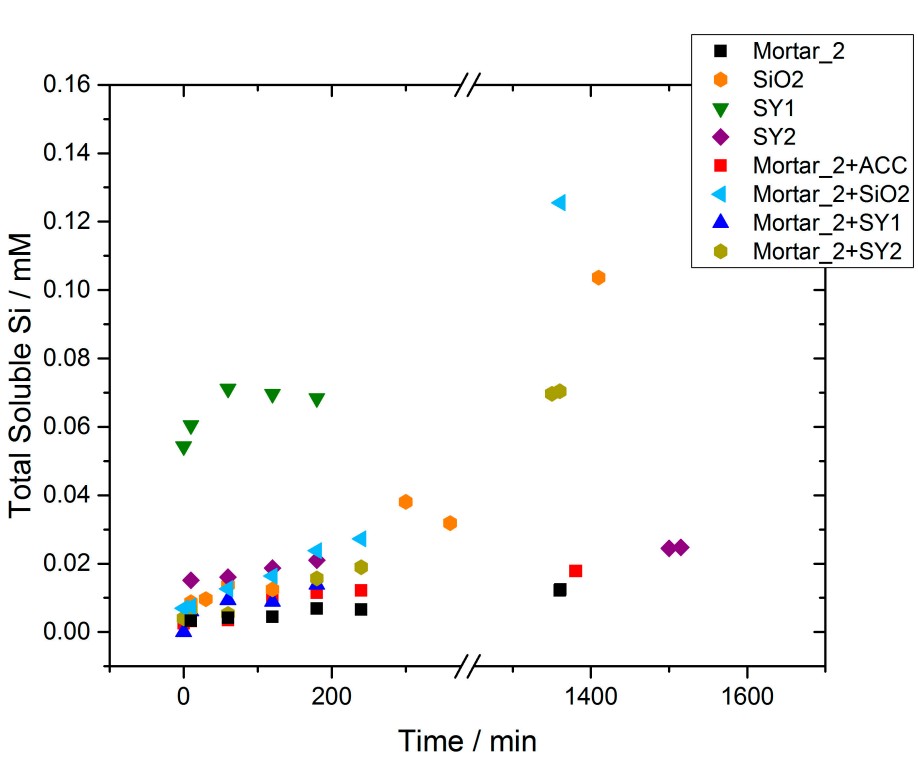

**(b)**

**Figure 10. (a)** Total calcium -time profile during the dissolution of mortar sample 2 in solutions undersaturated with respect to calcite at constant pH; ($\sigma_{calcite}$ = 0.89); 25 °C; pH 6.50; 0.1 M NaCl; (■)Mortar 2; (▼)ACC nanoparticles; (▼) SY1; (♦) SY2   nanoparticles; (▲) mortar 2 treated with SY1 nanoparticles; (●) Mortar 2 treated with SY2 nanoparticles . **(b)** Total silicate- time profile: (■) Mortar 2; (●) am-SiO2; (▼) SY1; (♦) SY2; (■) Mortar 2+ACC; (◄) Mortar 2+am-SiO$_2$; (▲) Mortar 2 + SY1; (●) Mortar 2 + SY2

The total calcium- time and total silicate-time profiles measured for the dissolution of mortar sample 3 (43% Ca/57% Si) in solutions undersaturated with respect to calcite at constant pH 6.50, are shown in the respective plots in Figure 11. The initial rates calculated for the dissolution of both untreated and treated with nanoparticles mortar sample 3 are summarized in Table 2. The treatment of the mortar with ACC nanoparticles, resulted in lower rates of dissolution in comparison with the corresponding to ACC nanoparticles, suggests that for this type of mortar (silica in excess) there is a strong interaction between the mortar and the ACC nanoparticles. Despite the fact that the measured rates were higher in comparison with the corresponding untreated mortar, in combination with the morphological examination of the mortar grains after dissolution, it was clear that the ACC nanoparticles functioned as sacrificial, dissolving at significantly lower rates in comparison with the free ACC nanoparticles, providing protection to the test historic mortar sample. Treatment with the silica particles resulted in rates of dissolution of the treated mortar, which were quite close to the corresponding am-SiO$_2$, suggesting that the treatment provides the mortar with sacrificial silica, thus protecting the mortar from dissolution. Apparently, in this case, the interaction between the deposited am-SiO$_2$ with the mortar rich in silica was not sufficiently strong to affect the performance of the treated mortar specimen with respect to dissolution in pH 6.50. The treatment of the mortar sample with SY1 and SY2 nanoparticles resulted in lower rates of dissolution of the mortar in comparison with the respective rates for each of the SY1, SY2 nanoparticles separately. Treatment with SY2 nanoparticles resulted in a significantly lower rate of dissolution at pH 6.50, in comparison with the rate of dissolution of SY2 nanoparticles. It may therefore be suggested that this reduction corresponded to stronger interactions of SY2 with the mortar. It is very likely that ACC is consolidated between the silica of the underlying mortar and the overlayer of am-SiO$_2$ of SY2. In the case of treatment with SY2 nanoparticles, it is possible that the higher silicate concentrations present in the undersaturated solutions reduce the rate of dissolution of the ACC.

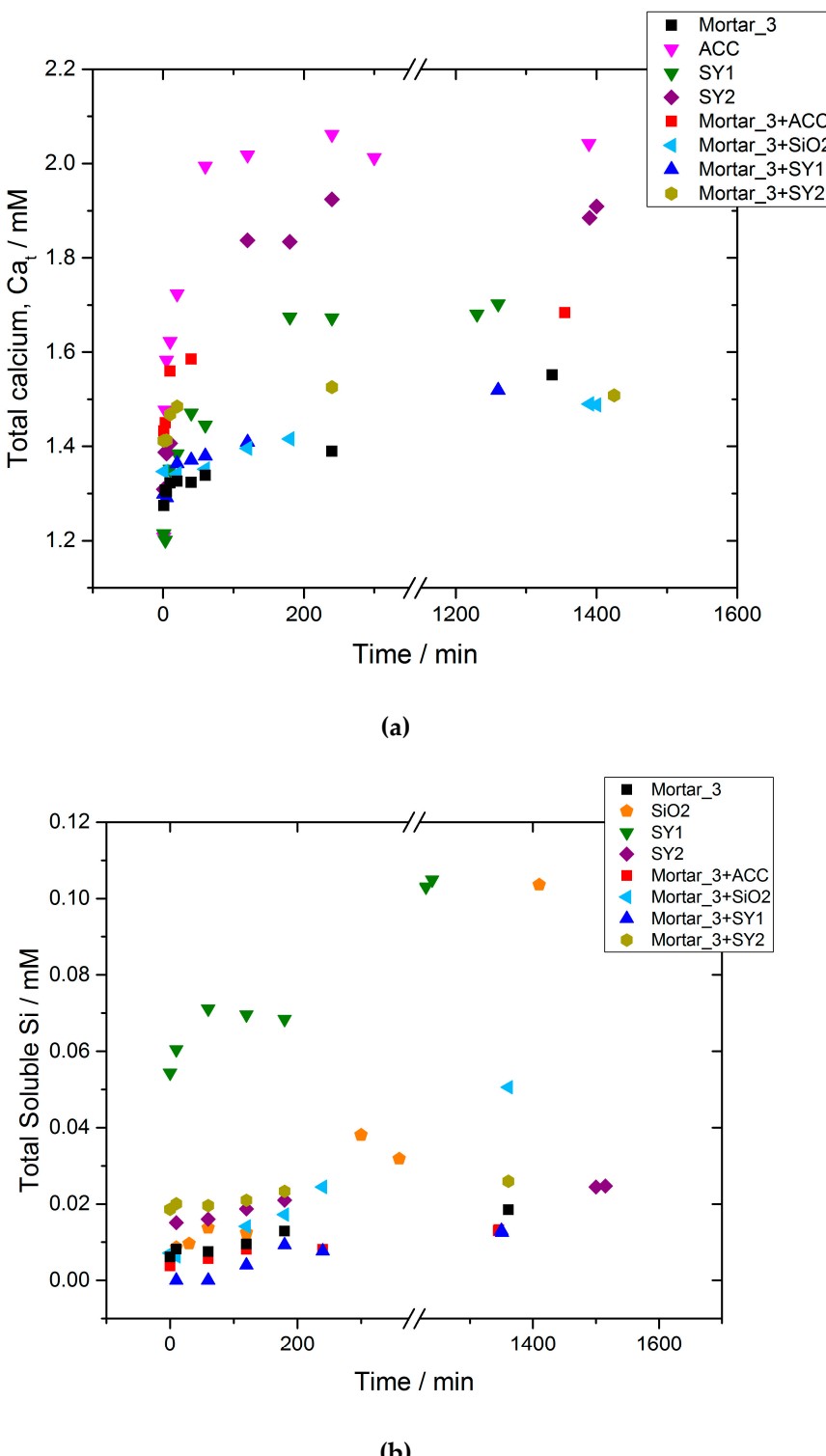

**(a)**

**(b)**

**Figure 11.** Total calcium-time profile measured for the dissolution of mortar sample 3, in solutions undersaturated with respect to calcite ($\sigma_{calcite}$ = 0.89) at constant pH; 25 ∘C; pH 6.50; 0.1 M NaCl; (■)Mortar 3; (■) Mortar 3 treated with ACC nanoparticles; (◄) Mortar 3 treated with am-SiO₂ nanoparticles; (▲) Mortar 3 treated with SY1 nanoparticles; (●) Mortar 3 treated with SY2 nanoparticles. (**b**) Total soluble silicate- time profile. Symbols as in (**a**).

**Table 2.** Initial rates of dissolution of the calcium carbonate component of nanoparticles of: ACC, SY1, SY2 and mortar samples 1,2,3 untreated and treated with ACC, am-SiO₂, SY1 and SY2 in

solutions undersaturated with respect to calcite, $\sigma_{calcite} = 0.89$, at constant pH 6.50; 25 °C, 0.1 M NaCl. Rates calculated per unit surface area of the dissolving sample.

| Material | Rate of Dissolution, $R_{diss}/\times 10^{-8}$ mol·m$^{-2}$·s$^{-1}$ | | |
|---|---|---|---|
| Mortar | 1 | 2 | 3 |
| | 3.8 | 9.9 | 10.0 |
| ACC | 170.0 | | |
| SY1 | 10.0 | | |
| SY2 | 43.0 | | |
| ACC Treated | 3.0 | 2.8 | 5.9 |
| am-SiO2 Treated | 0.5 | 1.3 | 0.2 |
| SY1 Treated | 3.0 | 5.6 | 6.9 |
| SY2 Treated | 0.9 | 3.2 | 6.2 |

The results of the measured rates of dissolution of the three mortar samples without (blank) and with treatment with nanoparticles, summarized in Table 2, are presented in Figure 12.

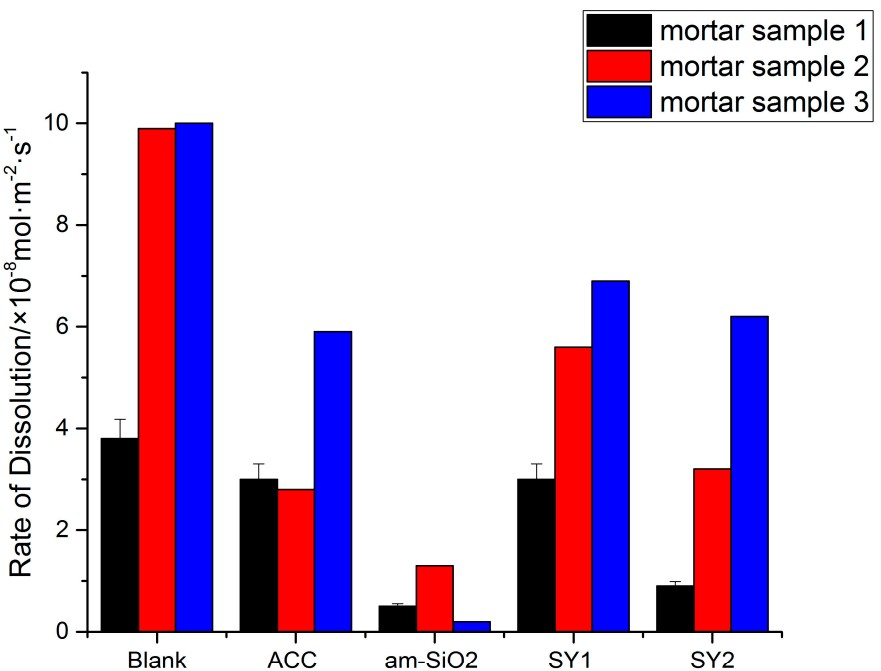

**Figure 12.** Rates of dissolution of historic mortar samples 1, 2 and 3 without (blank) and with treatment with ACC, am-SiO₂, SY1 and SY2 nanoparticles, in solutions undersaturated with respect to calcite ($\sigma_{calcite} = 0.89$) at constant pH 6.50; 0.10 M NaCl, 25 °C.

It is interesting to note that the historic mortars used in this study showed different behavior with respect to their dissolution in calcium carbonate solutions undersaturated with respect to calcite. Mortar sample 1, with the highest content in calcium carbonate, dissolved faster, while samples 2 and 3 with ca. 50% and 60% silica content dissolved faster. Since mortar sample 1 had the least porosity (Table 1) of all three samples, the higher calcium carbonate content in this sample corresponded to more complete consolidation of the respective mortar, which showed up in the slower rate of dissolution. The

treatment with ACC resulted in the reduction of the rates of dissolution, most probably because of the interaction of ACC with the silica content which consolidated the mortar. The decrease in the dissolution rate was more impressive for mortar samples 1 and 2. This suggestion was corroborated by the effect of the treatment of all mortar samples with am-$SiO_2$ nanoparticles, which because of their very small size are able to penetrate into the nanopores and consolidate the mortar samples. All mortar samples, upon treatment with am-$SiO_2$ particles, showed a drastic reduction of their rates of dissolution in the undersaturated acidic solutions. The effect was pronounced to the more porous material, mortar sample 3 (Table 1). The treatment of mortar samples with SY1 and SY2 nanoparticles did reduce significantly the dissolution rates of all three mortar samples for the treatment of which they were used. Clearly, SY2 treatment was more efficient. These types of nanoparticles were prepared by precipitation am-$SiO_2$ on ACC. The formation of layers of am-$SiO_2$ provided better consolidation. The highest effect (reduction of the rate of dissolution) on the calcite rich mortar sample 1 is due, on one hand, to the higher affinity of ACC for the calcitic component (61%), and to the consolidation effect of am-$SiO_2$ on the other.

## 4. Conclusions

Historic mortars consist mainly of calcite and silica. Deterioration due to weathering and more specifically to chemical dissolution due to wet precipitation can be greatly reduced by treatment with nanoparticles, which have both chemical and/or crystallographic affinity for the mortars. Strong bonding between the nanoparticles and the treatment matrix consolidated the deteriorated material, as shown by the morphological examination of the treated mortar samples. ACC, am-$SiO_2$ and composite nanoparticles prepared by the formation of ACC on am-$SiO_2$ nanoparticles, or the formation of am-$SiO_2$ on ACC, were shown to offer protection towards chemical dissolution in acid solutions (pH 6.50), undersaturated with respect to calcium carbonate. As shown from measurements of the rates of dissolution at constant pH 6.50, consolidation achieved through the interaction between silica and ACC is a key factor for the achievement of effective protection of historic materials from weathering. The nanoparticles deposited on the test historic mortars acted not only as consolidants, increasing the resistance of the material towards dissolution, but they also acted as sacrificial materials because of their apparent solubility. The novelty of the methodology proposed is in the use of ACC and am-$SiO_2$, and composites of these two components, at conditions in which they are stabilized for sufficiently long time periods, allowing for interaction with the mortar matrix treated. The treatment may be applied in situ (brushing or spraying) for the treatment of building elements and mortars.

**Author Contributions:** Conceptualization, C.P.; Methodology, E. I P., C.P. and P.G.K.; Validation, E. I P. and C.L.; Formal analysis, E.Z.; Investigation, E. I P. and C.L.; Data curation, E. I P. and E.Z.; Writing – original draft, E. I P.; Writing – review & editing, C.P. and P.G.K.; Supervision, C.P. and P.G.K.. All authors have read and agreed to the published version of the manuscript.

**Funding:** None

**Institutional Review Board Statement:** N/A

**Informed Consent Statement:** N/A

**Data Availability Statement:** N/A

**Conflicts of Interest:** The authors declare no conflict of interest.

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
