# Peer review of "Protection of Historical Mortars through Treatment with Suspensions of Nanoparticles"

_heritage, doi:10.3390/heritage6020064_

Round 1

Reviewer 1 Report

In the present work, the authors developed and applied suspensions of amorphous calcium carbonate (ACC), silicon oxide (am-SiO2) and composite nanoparticles by precipitation of ACC on am-SiO2 and vice versa. 

The following issues have to be addressed:

  • In all manuscript, keep the same format for pH 6.5, for temperature 25°C (e.g., lines 24, 102, 115), and for “am-SiO2” (e.g., lines 20 and 113, table 2)

  • At Introduction, present the novelty of this study in comparison with previous reported data in literature.

  • At chapter “2.1. Synthesis of nanoparticles”, specify the full name for all reagents. Specify the manufacturer, city, country for all reagents.

  • Replace Figure 2 with a clearer and original figure.

  • At line 276, authors mention “Composites of silica nanoparticles with ACC and silica particles.”. Number this sub-chapter.

  • At line 292, authors mention “Kinetics of dissolution”. Number this sub-chapter.

  • Remove “Mortar sample 2” (line 344) and “Mortar sample 3” (line 370), because in this chapter a comparison must be made between the samples and not separately.

English improvement is required. Some examples but not all are as the following:

Page 3, line 126, modify 0.005 dm3 ammonia (NH3, 32% w/v)  with “0.005 dm3 ammonia (NH3, 32% w/v)

Page 6, line 212, modify350nm)” with “350 nm

Author Response

Your review effort is appreciated.  the paper was improved thanks to your critical evaluation and comments.  Detailed response is provided in a separate document.

Reviewer 2 Report

In this study, suspensions of amorphous calcium carbonate (ACC), silicon oxide (am-SiO2) and composite nanoparticles were developed by precipitation of ACC on am-SiO2 and used to preserve the integrity of historical artifacts. The subject of the project is very interesting and the project method and results are given in an order.

1.      Figure 2 needs to be updated to be more descriptive. Chemical structures and reaction should be named.

2.      It is necessary to have scaling bars on SEM images. In addition, it may be more explanatory to write the details about the analysis.

3.      Error bars should also be added to the graphs.

4.      There are shifts in some columns in Table 2, it should be updated.

5.      The conclusion part can be expanded by considering the feasibility, originality and usefulness of the developed study.

6.      It is recommended to update the literature used within the scope of the study, especially to cover the last 5 years.

Author Response

(The authors gave the same response as above.)

Round 2

Reviewer 1 Report

Dear Sirs,

The manuscript was improved and it can be published in this form.